# Show, Don't Tell: Evaluating Large Language Models Beyond Textual Understanding with ChildPlay

## Abstract

The evaluation of Large Language Models (LLMs) often focuses on linguistic tasks, yet such assessments may not fully capture the models' general reasoning capabilities. We explore the hypothesis that LLMs, such as GPT-3.5 and GPT-4, possess broader cognitive functions, particularly in non-linguistic domains. Our approach extends beyond standard linguistic benchmarks by incorporating games like Tic-Tac-Toe, Connect Four, and Battleship, encoded via ASCII, to assess strategic thinking and decision-making. To evaluate the models' ability to generalize beyond their training data, we introduce two additional games. The first game, LEGO Connect Language (LCL), tests the models' capacity to understand spatial logic and follow assembly instructions. The second game, the game of shapes, challenges the models to identify shapes represented by 1s within a matrix of zeros, further testing their spatial reasoning skills. This "show, don't tell" strategy uses games to potentially reveal cognitive capabilities rather than simply querying the models. Our results indicate that despite their proficiency on standard benchmarks and temperature settings, GPT-3.5 and GPT-4's abilities to play and reason about fully observable games without pre-training is mediocre. Both models fail to anticipate losing moves in Tic-Tac-Toe and Connect Four, and they are unable to play Battleship correctly. While GPT-4 shows some success in the game of shapes, both models struggle with the assembly tasks presented in the LCL game. These results suggest that while LLMs like the GPT models can emulate conversational proficiency and basic rule comprehension, their performance in strategic gameplay and spatial reasoning tasks is limited in cognitive flexibility and generalization. Importantly, this reveals a blind spot in current LLM benchmarks that we highlight with our gameplay benchmark suite ChildPlay (GitHub Repository). Our findings provide a cautionary tale about claims of emergent intelligence and reasoning capabilities of LLMs that are roughly the size of GPT-3.5 and GPT-4.

## 1  Introduction

Typically, LLMs are transformer-based models that process input text and generate output text in a coherent and contextually appropriate manner. They utilize the self-attention mechanism to weigh the importance of different words in a sentence relative to each other [33, 6]. Input text is tokenized, converted into vectors using embeddings, and processed through transformer layers that calculate attention scores to dictate focus on relevant tokens [33, 6, 12]. The model then selects the next token based on learned distributions, iteratively generating an arbitrarily long sequence of text [33, 6, 12]. With their enormous parameter counts, from Alpaca with 7 billion parameters [29], to LLaMA with 65 billion [31] or even PaLM and its 540 billion parameters [11], these neural networks have learned to model complex linguistic abstractions, capturing patterns in syntax, semantics, pragmatics, and even elements of style and tone [6, 7, 21].

Submitted to 38th Conference on Neural Information Processing Systems (NeurIPS 2024). Do not distribute.

Benchmarks for evaluating Large Language Models (LLMs) have been designed to assess comprehension, generation, and adaptability across a wide range of language tasks. Datasets like SQuAD, GLUE, BIG-bench, and the lm-evaluation-harness offer various test types, including multiple-choice questions, reading comprehension exercises, and dialogue completion tasks. These benchmarks deploy metrics such as response correctness, language generation fluency, and the ability to maintain contextually relevant dialogue [22, 34, 2, 26]. Other benchmarks like SuperGLUE, ANLI, TruthfulQA, and HellaSwag have been developed to evaluate different aspects of LLM performance, such as natural language understanding, commonsense reasoning, and factual knowledge about diverse topics [35, 20, 18, 37].

Recent studies have explored alternative approaches to evaluate LLMs' reasoning abilities in non-linguistic modalities. Liga and Pasetto modeled the game Tic-Tac-Toe using ASCII characters, pitting LLMs against the minimax algorithm to observe emergent features, which, according to the authors, might be akin to consciousness. The minimax algorithm is widely considered the optimal algorithm for playing tic-tac-toe, as it guarantees a win or draw against a perfect opponent [27, 1]. While LLMs performed well in some instances, they generally failed to win against the minimax algorithm, often resulting in a draw [17]. Topsakal and Harper [30] used Tic-Tac-Toe encoded with list and illustration prompts in their study. They found that while GPT-4 secured the most wins, it did not always win, indicating that GPT models cannot play Tic-Tac-Toe optimally. This contradiction raises the question: can we truly say the model knows how to play Tic-Tac-Toe if it can explain optimal strategies (see Appendix A.3) but does not consistently win? Or is its performance merely the result of probabilistic outcomes?

Some critical studies have highlighted the need for caution in interpreting LLMs' capabilities through benchmarking. Lappin et al. assessed their strengths and weaknesses, finding that they excel at many language tasks but struggle with deeper reasoning, world knowledge integration, and context understanding beyond local co-occurrences [16]. And Zečević et al. argued that LLMs may discuss causality but lack true causal reasoning based on interventions and counterfactuals [38].

Bender et al. argue that the lack of transparency and potential risks associated with these large, opaque models raise concerns about their trustworthiness and accountability [3]. While the criticism of Bender et al. focuses on the social dimension of the problem of interpretability and trustworthiness, recent work by Schaeffer et al. critics emergent capabilities and the perceived intelligence of LLMs. They suggest that some claimed "emergent abilities" of LLMs may be an artifact of the choice of evaluation metric, rather than fundamental changes in model behavior [23]. Their analyses demonstrate how the use of nonlinear or discontinuous evaluation metrics can create the illusion of emergent abilities, even when the underlying model performance changes smoothly and predictably with scale.

This critique of the evaluation metrics used in assessing LLMs invites a deeper exploration of general intelligence - specifically how it can be reliably measured and observed in AI through rigorous and realistic tests that extend beyond linguistic prowess to include broader cognitive functions. If we must define general intelligence (GI), one is to use the "g factor," which refers to the ability to reason, plan, solve problems, think abstractly, and learn quickly across a wide range of domains [24, 4, 36, 9, 8]. GI then involves higher-order cognitive processes that go beyond specific skills or knowledge domains [14, 15].

A critical issue that arises in analysing the reasoning capabilities of large and opaque models like the GPT series, is training-test set cross-contamination, which becomes increasingly problematic for the most advanced models [6]. The massive training datasets used, comprising extensive portions of the internet, are often untraceable and completely anonymous to researchers outside the initial developer groups, to some extent even to the developers themselves, making replication studies impossible [6, 13]. The exact amount and identity of data used to train models like GPT-3.5 or GPT-4 has not been publicly disclosed, posing a risk of rendering current benchmarking efforts meaningless due to cross-contamination.

Researchers have attempted to counter the contamination problem using N-Gram Overlap as a metric for detection, by eliminating or withholding results for tests where answers were present in the training data [6]. However, this method has been criticized. Blodgett et al. point out, for example, that such heuristic approaches to mitigating biases in NLP systems can be problematic and may not fully address the underlying challenges [5]. The method is also limited in that it fails to consider the context in which N-Grams appear and may discount synonymous or analogous text worded

94 differently. Additionally, the decision to use a 200-character window around detected N-Grams is
95 arbitrary and may not accurately reflect the influence of surrounding text on model learning.

96 In this work we introduce ChildPlay, a suite of non-language-based games like Tic-Tac-Toe, Connect-
97 Four, Battleship, LEGO Connect Language, and the game of Shapes, to assess reasoning, strategic
98 capabilities, symbolic reasoning, and pattern recognition abilities of large language models (LLMs)
99 beyond traditional linguistic modalities. Games provide structured environments with clear success
100 criteria, making them suitable for evaluating strategic thinking, planning, and long-term decision-
101 making of LLMs [25, 17, 30]. Their dynamic and adversarial nature resembles real-world scenarios,
102 assessing generalized intelligence and reasoning beyond the training domain [25, 17, 30]. We encode
103 these games using ASCII representations to minimize dataset contamination issues prevalent in
104 contemporary LLM benchmarks [6, 17].

## 2   Experiments

106 Specific tasks in the BIG-bench benchmark [2], among others, are categorized as either zero-shot,
107 one-shot, or multi-shot [6]. Our tasks fit the zero-shot category, as models are given only a brief
108 explanation at inference time with no examples for playing beyond the explained formalism. To
109 demonstrate the reasoning capabilities of LLMs beyond their training data, we focus on a modality not
110 explicitly trained for: spatial reasoning about ASCII sequences. An agent capable of true abstraction
111 should be able to encode and interpret these sequences if the rules are explained or known.

112 For this purpose, we developed several tasks, including LEGO assembly, ASCII games of Tic-Tac-
113 Toe, Connect-Four, and Battleship, as well as identifying simple geometrical shapes represented as 1s
114 in 15-sided grids of 0s. The same models were deployed over all experiments, namely *gpt-3.5-turbo-*
115 *1106*, and *gpt-4-1106-preview*, which in this paper are referred to as GPT-3.5 and GPT-4, respectively.
116 Every experiment was tested across different temperature settings (t) per model, namely t=0, t=0.5,
117 t=1, and t=1.5. When asked about their understanding of the tasks, GPT-3.5 and GPT-4 were able to
118 generate board states and explain the queried games, including their rules and optimal play. Thus, we
119 consider the tests valid: if the models are truly capable of reasoning, they should be able to play these
120 games optimally given that they "know" and are capable of explaining what playing optimally means
121 (see Appendix A.3). Experiments ran over night, at minimum taking a couple of minutes and at most
122 taking a few hours.

**Lego Connect Language (LCL)**   We invented a formal language we call LEGO Connect Language
124 (LCL). More specifically, we propose $LCL_2$ as a language to instruct assembly in 2D on the x and y
125 axis (this can easily be generalised to $LCL_3$ - instructions along the x, y, and z axis). The models
126 were given instructions and their output was fed through a visualizer script to generate the images
127 contained in this work. Only 2x4 pieces were allowed. A piece $P$ (see Fig 1) is then defined as a
128 tuple $P = (l, w, (x, y), c, h)$. A construction, $M$, is then a valid construction in $LCL_2$ *if* no pieces
129 are overlapping and all pieces are connected to other pieces. Namely, a Lego piece is connected
130 through interlocking pegs, not by merely touching sides. And secondly, two Lego pieces overlap
131 when they share the same y-coordinate and any part of their length has the same x-coordinate.

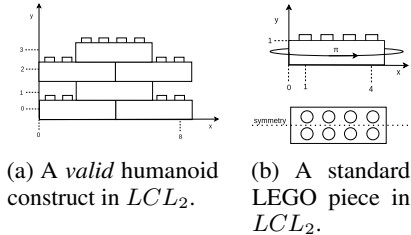

(a) A *valid* humanoid
construct in $LCL_2$.

(b) A   standard
LEGO   piece   in
$LCL_2$.

Figure 1: Introducing $LCL_2$.

**Game 1: Validity Testing**   In this experiment, we evaluate the ability of different models to validate
133 the correctness of a given Lego construct. The constructs are generated to be either valid or invalid.
134 A construct is considered valid if there is no horizontal overlap between pieces, and pieces must
135 connect via overlapping pegs such that the whole assembly is connected (no floating pieces). The

models, namely GPT-4 and GPT-3.5, are then tasked with predicting the validity of each construct. The evaluation metric for this experiment was the proportion of correct validations, measured across different temperature settings.

**Game 2: Construct Generation**    In this experiment, the models attempt to generate valid LCL constructs. Each construct description consists of a list of tuples, where each tuple specifies the coordinates and color of a Lego piece. The models generated these constructs based on prompts and the validity of the constructs was automatically evaluated. The metric for this experiment was the proportion of valid constructs generated, measured across different temperature settings.

We automatically produced 800 images for the validity test, half valid and half invalid ones. Then each model was queried to produce 100 images at each temperature setting, which we then checked for validity. We believe our use of LCL is related to the tests found in Bubeck et al. [7], where JavaScript or LaTeX was used to prompt GPT-4 to produce images. However, while the images in Bubeck et al. [7] included common examples such as letters, a car, a truck, a cat, a dog, a person, a pig, a house, and a unicorn, all of which are likely represented in the training data in JavaScript or LaTeX, LCL challenges the model to step outside of its learned data distributions by remaining abstract.

**Three Board Games: Tic-tac-toe, Connect-four, and Battleship**    In the case of the three board games, each new board state was accompanied by the introductory game explanation sent through the OpenAI API in a zero-context testing environment. The models were provided with the current board state and an opponent making moves at random, with the LLM always playing as the first player, which is advantageous in all three games. Context beyond the initial instruction and the current board state was deemed irrelevant since these games are fully observable, meaning every board state contains all the necessary information to play optimally. The input to the game was simply two scalars for the row-column pair or just a scalar for the column number in the case of connect-four.

For the battleship game, ships ('S') were randomly initialized, always horizontally, with varying sizes spanning between 2 and 5 cells. When there is a hit by either player, the position is marked with an 'X' on both players' boards. If the guess was a miss, an 'O' is placed on the position instead.

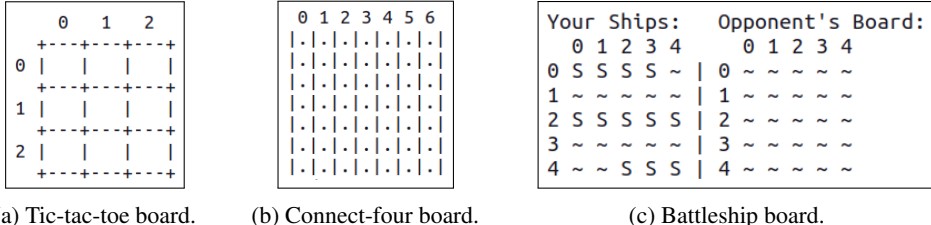

(a) Tic-tac-toe board.        (b) Connect-four board.            (c) Battleship board.

Figure 2: Initial board states as presented to the LLM (the ship positions in the Battleship board are randomised with every initialisation, including ship length).

**The Game of Shapes**    In the case of the game of shapes, preliminary work involved probing the models to determine what geometric shapes they consider basic by prompting them multiple times. The first three shapes consistently mentioned were square, circle, and triangle (not necessarily in that order). The game then consists of finding a basic geometric shape "hidden" behind 1s within a matrix of 0s in a multiple-choice fashion. Four shapes were used as options: the circle, the rectangle, the triangle, and the cross, but only the latter three were ever shown to the model (cf. Fig. 3).

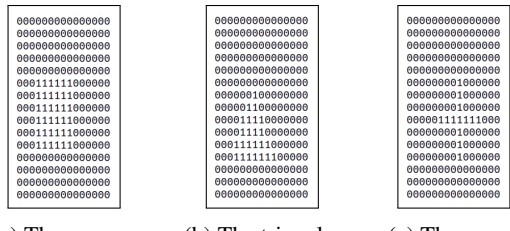

(a) The square.    (b) The triangle.    (c) The cross.

Figure 3: Matrices containing shapes used in the game of Shapes.

# 3   Results

As previously stated, Tic-Tac-Toe as a benchmark has been tackled before [17, 30]. Since it is quite popular, we decided to replicate it before creating new games. But this time using an ASCII encoding instead of a list of moves such that we can gauge spatial reasoning through symbolic reasoning. For comparison with the model's performance, Fig. 4 presents the Tic-Tac-Toe match results of the *minimax* algorithm against the same random player the models played against. This outcome creates a baseline for optimal play against a random player.

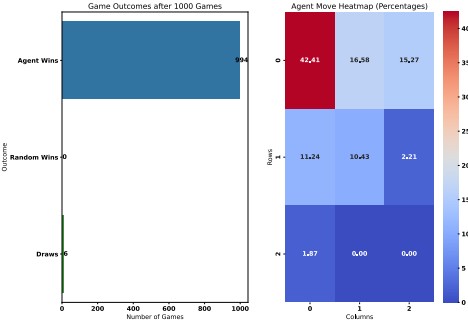

Figure 4: Minimax vs random player.

**Tic-tac-toe, Connect-four, and Battleship**    To check for a win, we determine if the player has successfully connected the winning number of pieces in a row on the board, which could be horizontally, vertically, or diagonally. To detect missed and blocking moves, we simulate all potential moves for the player by checking if placing a piece in any column leads to a win. If such a move is found, and the player does not execute it on their turn, it is recorded as a missed win, if such a move is found for the opponent and the player does not block it, we register it as missed blocking move. We define *incorrect moves* to mean a move that was illegal, such as playing a position that has already been played. This results in an immediate loss.

Fig. 5 encompasses comparative results from playing Connect-Four, Tic-Tac-Toe, and Battleship. Each subfigure, 5a, 5b, and 14, respectively, outlines the number of games won by the models.

Unfortunately, the models were incapable of following the rules for the Battleship game, that is, regardless of temperature, the models lose the large majority of games, with GPT-4 not winning a single game due to incorrect moves (cf. Fig. 16). GPT-3.5 wins around 10% of the matches at low temperatures, but none at higher temperatures, we refer to Fig. 14 in the Appendix A.1.3 instead.

It is notable that both GPT-3.5 and GPT-4 exhibit their poorest performance in both Connect-Four and Tic-Tac-Toe at a temperature setting of 0, indicative of deterministic play that reflects the models' learned strategies (Appendix A.1). The Random Player's normal distribution across columns (Fig. 12) suggests a lower likelihood of countering GPT's central strategies, in both games, but particularly at Tic-Tac-Toe where GPT-3.5 commits more errors than GPT-4, significantly impacting outcomes due to incorrect moves (Fig. 5b). These errors generally increase with temperature, probably due to enhanced choice randomness (Fig. 10). This explains the lack of direct model losses from final defeating moves since losses often result from illegal moves.

Average game moves, missed wins, and blocks in both Tic-Tac-Toe and Connect-Four are further illustrated in Figs. 6a and 6b, highlighting a decrease in these metrics as temperature rises, suggesting that higher settings potentially broaden the explored moves within the models' strategies. Conclusively, neither model plays the games optimally, as evidenced by the considerable number of missed wins and blocks. Both subfigures demonstrate that, as temperature increases, the number of missed wins and blocks decreases. This might suggest that higher temperature settings potentially increase the explored moves in the models' learned strategy, in case there is any. We can conclude the same as before, namely that neither model can play Tic-Tac-Toe optimally given the number of missed wins and missed blocks.

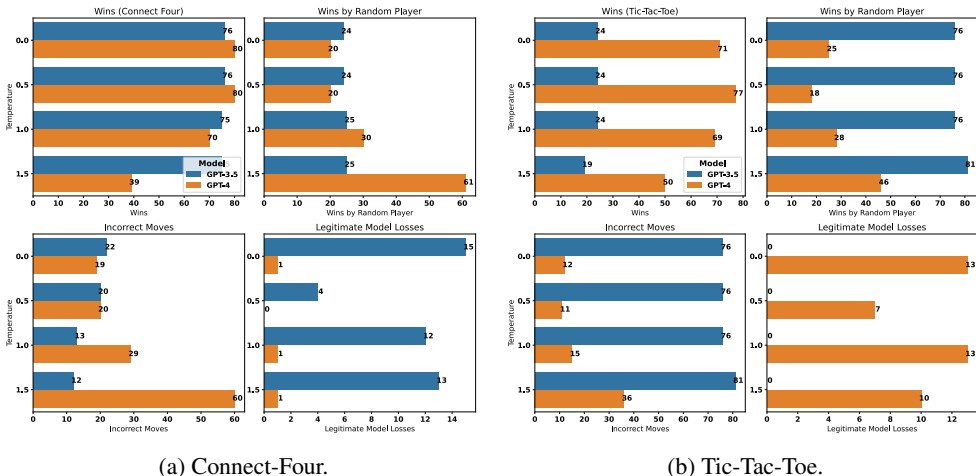

(a) Connect-Four.                                    (b) Tic-Tac-Toe.

Figure 5: Incorrect Moves, Wins, and Losses Per Player in the Three Board Games.

The number of moves of GPT-3.5 and GPT-4 per game can be thought of as a measurement of stability in gameplay, not just against the random player, but in general, given that a longer game entails that the model is not losing to illegal moves or to its oponnent. It increases linearly with temperature, inversely correlated with performance measured by the decrease in missed wins and blocks. Tic-Tac-Toe shows a linear improvement, whereas Connect-Four experiences an exponential boost in performance from temperature 0 to 0.5, followed by a linear decline. The random player consistently performs better against GPT-3.5 in Tic-Tac-Toe but loses more frequently in Connect-Four. Both models struggle with blocking or seizing winning moves from the random player. An analysis of the move heatmaps (cf. Appendix A.1) explains why winning Connect-Four against a random player appears straightforward. As the model consistently places pieces in the same column, the probability of the random player losing increases with the board size. However, even under these challenging conditions, the random player still secures wins in at least 20% of the games played against GPT-4.

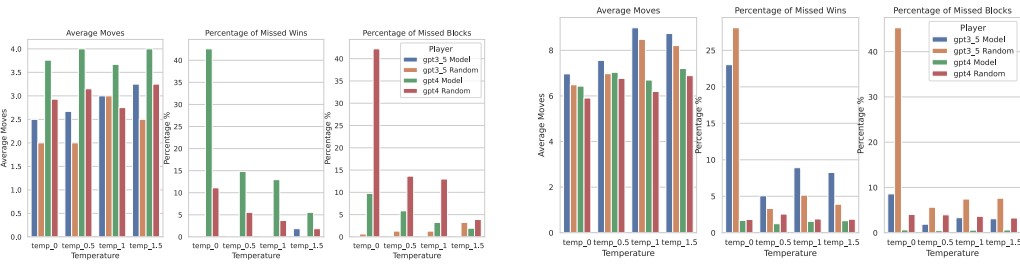

(a) Tic-tac-toe: Missed Wins and blocks.          (b) Connect-Four: Missed Wins and blocks.

Figure 6: Average Moves, missed wins, and missed blocks for Tic-tac-toe and Connect-Four.

**Shapes**    In the game of Shapes, a correct detection happens when the player's selected shape corresponds with the shape shown on the board. Players have four choices: "circle," "triangle," "square," and "cross". Notably, a circle is never actually displayed to the model, and the positions of these choices are not randomized to test if the model displays any inherent bias for the question order. This does not affect the outcome, since the game does not change across different sessions as it is designed to operate within a single question-response framework.

In the shape detection tests, GPT-3.5's performance was approximately equivalent to random chance when identifying triangles and crosses, yet it completely failed to recognize squares. In contrast, GPT-4 performed remarkably well, successfully identifying shapes with an accuracy of 80% or higher, particularly proficient at recognizing triangles[1].

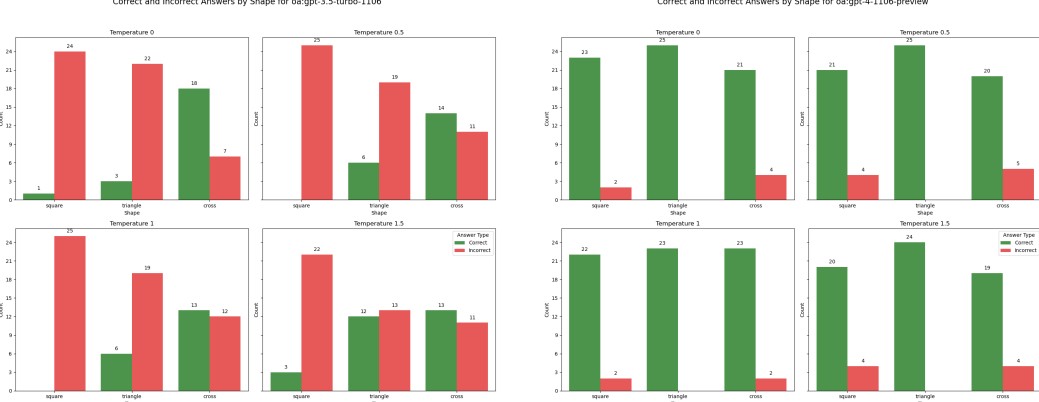

(a) Results for the Shapes game, as played by GPT-3.5.

(b) Results for the Shapes game, as played by GPT-4.

Figure 7: Experiment results for the Shapes game, comparing GPT-3.5 and GPT-4.

**LCL**    In the game of LCL, both models systematically failed to respect the two rules, namely that Lego pieces must be connected through interlocking pegs, not by merely touching sides, and secondly, that no Lego pieces may overlap, which occurs when they share the same y-coordinate and any part of their length has the same x-coordinate. For example, Figs. 8, 8a, and 8b show valid LCL assemblies, while Figs. 8c and 8d show invalid LCL structures. Figs. 8a and 8b show valid LCL assemblies, while subfigs. 8e and 8g show invalid output from GPT-3.5 generated at temperature 0. While Fig.8f shows a valid output from GPT-4 at temperature 1.5. Other images (Figs. 8i, 8j, 8k, and 8l) are of invalid output[2].

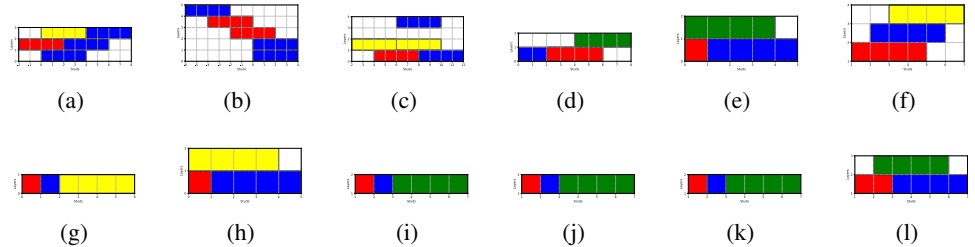

(a)       (b)       (c)       (d)       (e)       (f)

(g)       (h)       (i)       (j)       (k)       (l)

Figure 8: Structures automatically generated for the LCL validity test and structures generated by GPT-3.5 and GPT-4 for the construction generation test.[3]

---

[1]At higher temperatures, some of GPT-4's responses were discarded by our parser when the model generated invalid Unicode output, and thus were not included in the final evaluation. This discrepancy is evident in Fig. 7b, for instance, where the sum of correct and incorrect choices does not total 25 at temperatures 1 and 1.5.

[2]Fig. 8i = GPT-4 at temperature 0, Fig. 8j = GPT-4 at temperature 0.5, Fig. 8k = GPT-4 at temperature 1, and Fig. 8l = GPT-4 at temperature 1.5.

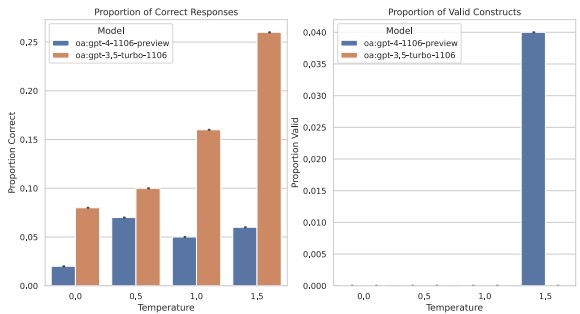

Figure 9: LCL results after 100 runs with 50/50 valid/invalid examples for the validity test and 100 experiments per temperature per model for the construction modality using 3 pieces.

Fig. 9 shows a roughly linear increase in the proportion of correct answers during the validity test as a function of temperature. However, only GPT-4 produced a small minority of valid LCL constructs (namely 0.04 of a total of 400 = 16), while GPT-3.5 did not manage to produce a single valid LCL construct.

## 4 Discussion

In Tic-Tac-Toe, both models underperform relative to the minimax algorithm baseline, while showing mixed performance at Connect-Four. GPT-4 performs unexpectedly well at the Shapes game, but GPT-3.5 does very poorly. Also unexpectedly, both models fail to assemble or detect valid Lego structures in the LCL game. In Battleship, the models' failure to follow game rules, especially at higher temperature settings, indicates a significant limitation in their ability to understand and apply structured game rules. The linear increase in the number of moves with temperature suggests that higher temperatures lead to greater exploration of possible moves, but do not improve strategic performance. The increase in missed wins and blocks with temperature further supports this, as greater randomness in decision-making does not enhance the models' strategic play.

Overall, these results show that while GPT-3.5 and GPT-4 can play simple games to some extent, they struggle with more complex tasks and do not consistently apply optimal strategies. The performance gap between the models and the minimax algorithm highlights the limitations of current language models in tasks requiring precise strategic reasoning and the failure to play Battleship and LCL demonstrates a failure in rule adherence.

The primary aim of contemporary benchmarks for LLMs has been to assess these models through adaptations of Turing's test [32], evaluating their capability to process and respond to language inputs comparably to humans. However, defining the language problem solely in these terms may overlook deeper complexities. While the transformer architecture in deep neural networks has enabled models smaller than GPT-4 to exhibit what Wilhelm von Humboldt described as the "infinite use of finite means" [19] or their ability to generate a potentially unlimited number of contextually relevant sentences [28] (an idea popularised by Chomsky [10]), this does not necessarily imply that these models have mastered a form of reasoning. Rather, they may simply be engaging in an advanced form of pattern imitation.

### 4.1 Limitations and Future Work

Our proposed benchmark, ChildPlay, primarily uses binary (win/loss) outcomes for games, which can be considered discontinuous metrics. Mathematically, these are expressed as:

$$\text{Metric}(x) = \begin{cases} 1 & \text{if win} \\ 0 & \text{if loss} \end{cases}$$

---

[3]Images in Fig. 8 were not directly produced by the GPT models. Instead, the formal descriptions of these images were generated by the models and subsequently passed to a script for rendering available in the GitHub Repository.

This formulation may exaggerate perceived capabilities by registering a full loss even if the model's failure was marginal. We try to avoid this simplistic classification by registering, for example, the choice of moves on the board games (see Appendix A.1) as well as the count of missed blocks and missed wins (cf. Fig. 6). In contrast, tasks involving shape recognition or LCL could utilize more continuous metrics, providing a smoother performance gradient and potentially more accurate reflections of a model's reasoning abilities.

Using discontinuous metrics in strategic games could manifest as sharp transitions in model evaluation:

$$\text{Performance}(N) = \delta(\text{outcome}_N - \text{threshold})$$

where $\delta$ is the Dirac delta function, accentuating a sudden jump in perceived ability when the model first succeeds. Nonlinear metrics in the shape game or LCL tasks may not exhibit such abrupt transitions but could still misrepresent gradual improvements:

$$\text{Performance}(N) \approx \exp(-\alpha N^{\beta})$$

where $\alpha > 0$ and $\beta < 0$ dictate the rate of improvement. This expression reflects smoother but potentially misleadingly slow progress.

Based on the perspective from Schaeffer et al. [23], one could argue that the games proposed in ChildPlay may not entirely reflect true generalization or emergent abilities. If these benchmarks are akin to nonlinear or discontinuous metrics, they might exaggerate the weaknesses or strengths of LLMs in strategic games. For instance, a sharp failure in a game like Tic-Tac-Toe might not mean the model lacks strategic reasoning universally but that it fails under the specific discontinuous conditions of the game setup, or of temperature. Such an assessment could lead to the erroneous conclusion that LLMs are generally poor at strategic decision-making when, in fact, they might only be unsuited to the specific scenarios or metrics used in ChildPlay.

Conversely, unlike continuous metrics that might smooth over deficiencies and give a misleading picture of gradual improvement, the use of clear, structured games as benchmarks could provide a direct assessment of an LLM's cognitive and strategic abilities regardless of metric continuity. That is, given that the model has not been overfitted on the game.

# 5 Conclusions

Non-language-based tasks are important as they challenge models to demonstrate generalization across different information encodings or forms of input, and, most importantly, to actually delve into out-of-training-distribution topologies. Testing LLMs like GPT-4 (according to OpenAI, the current contender to AGI [7]) beyond the text they were primarily trained on via our "show, don't tell" strategy, we demonstrate that it is still mediocre at best at very simple reasoning tasks that are outside of its training data. The models fail to play optimally at very simple games, such as tic-tac-toe, battleship, and connect-four. We also experimented with LEGO assembly, finding the LLMs still performing poorly. Mixed results were found at the task of interpreting geometric shapes from binary grids. These tasks are then designed to test reasoning without relying on language skills, such that the model cannot get by through parroting - it must be capable of playing the game. In the context of BIG-bench, our tasks would fit in the "non-language" category. Currently, this category shows 16 active tasks, including some explicit ASCII recognition tasks, chess, and Sudoku, however, to the best of our knowledge, no task like ours [2]. Hence, we believe that ChildPlay is a useful addition to the suite of current established LLM benchmarks.

In general, this work is relevant in that developing games allows us to critically examine claims regarding a models' ability to perform reasoning and problem solving regardless of the persistent problem of data contamination. In other words, we explore what the model knows by making it play games instead of asking it how to play them. Our results suggest that current LLMs show disappointing performance in terms of problem solving capabilities and reveal important aspects to be considered for future improvements.

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

# A Appendix / supplemental material

## A.1 Move Mapping

### A.1.1 Tic-Tac-Toe

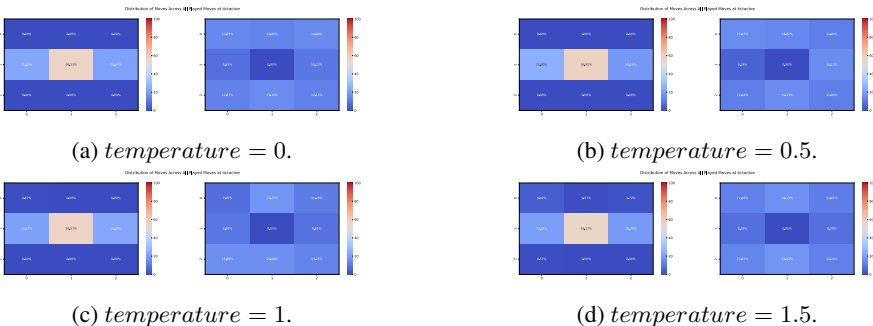

(a) $temperature = 0.$

(b) $temperature = 0.5.$

(c) $temperature = 1.$

(d) $temperature = 1.5.$

Figure 10: Heatmap of model GPT-3.5's moves for the tic-tac-toe game.

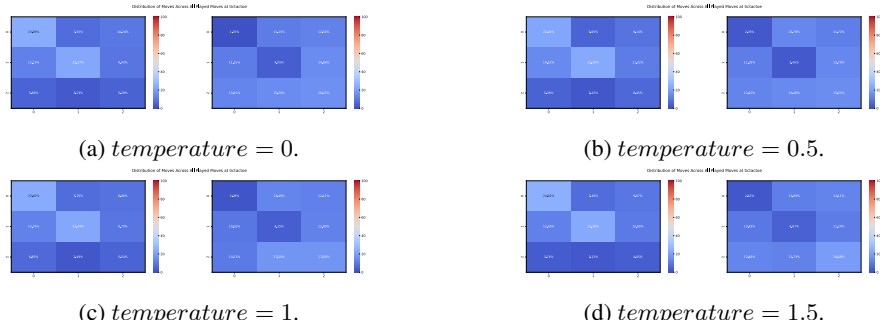

(a) $temperature = 0.$

(b) $temperature = 0.5.$

(c) $temperature = 1.$

(d) $temperature = 1.5.$

Figure 11: Heatmap of model GPT-4's moves for the tic-tac-toe game.

### A.1.2 Connect-Four

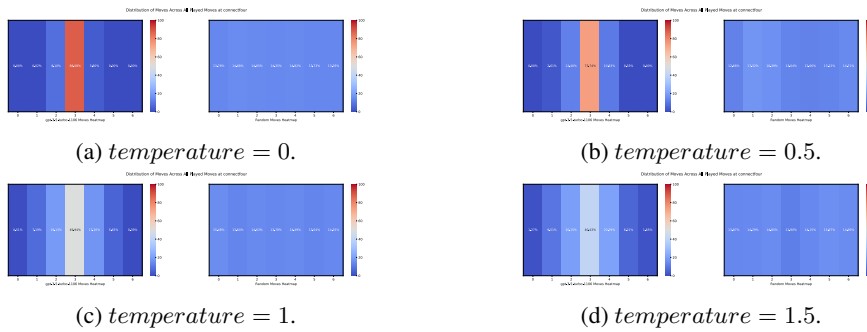

(a) $temperature = 0.$

(b) $temperature = 0.5.$

(c) $temperature = 1.$

(d) $temperature = 1.5.$

Figure 12: Heatmap of model GPT-3.5's moves for the connect-four game.

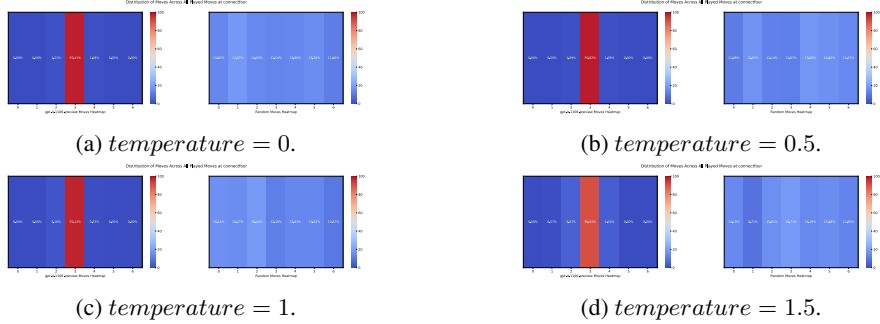

(a) $temperature = 0$.

(b) $temperature = 0.5$.

(c) $temperature = 1$.

(d) $temperature = 1.5$.

Figure 13: Heatmap of model GPT-4's moves for the connect-four game.

### A.1.3 Battleship

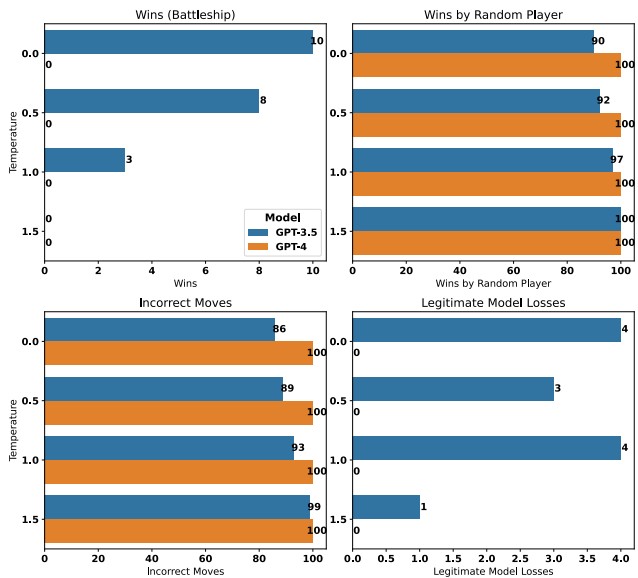

Figure 14: Battleship.

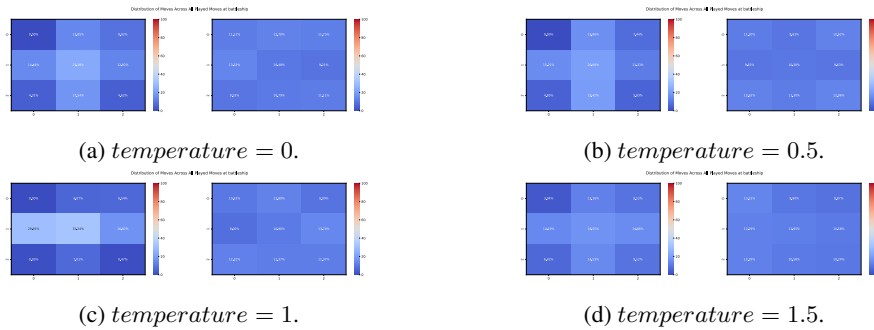

(a) $temperature = 0$.

(b) $temperature = 0.5$.

(c) $temperature = 1$.

(d) $temperature = 1.5$.

Figure 15: Heatmap of model GPT-3.5's moves for the battleship game.

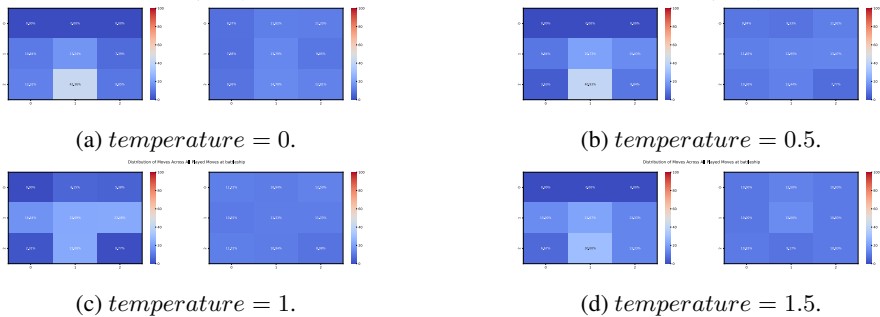

(a) $temperature = 0$.

(b) $temperature = 0.5$.

(c) $temperature = 1$.

(d) $temperature = 1.5$.

Figure 16: Heatmap of model GPT-4's moves for the battleship game.

## A.2 Shapes

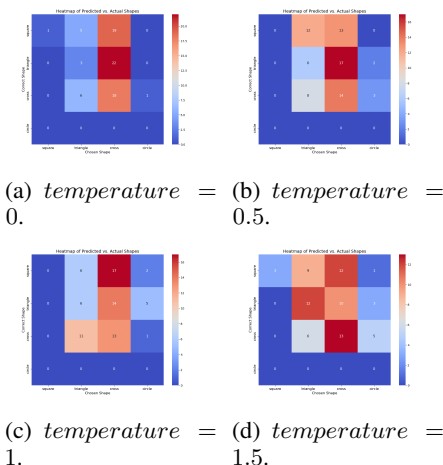

(a) $temperature$ $=$ 0.

(b) $temperature$ $=$ 0.5.

(c) $temperature$ $=$ 1.

(d) $temperature$ $=$ 1.5.

Figure 17: Heatmap of model GPT-3.5's moves for the shapes game.

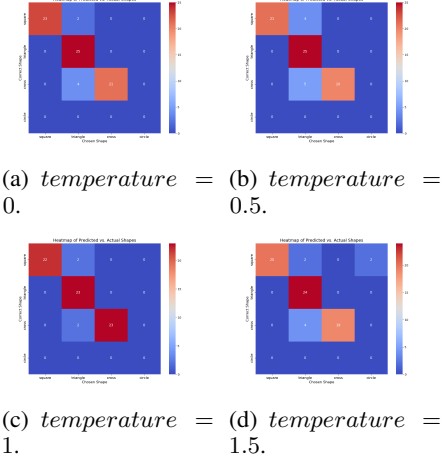

(a) $temperature$ $=$ 0.

(b) $temperature$ $=$ 0.5.

(c) $temperature$ $=$ 1.

(d) $temperature$ $=$ 1.5.

Figure 18: Heatmap of model GPT-4's moves for the shapes game.

 **A.3 Prompting GPT About Optimal Play**

| Game | Explanation |
|---|---|
| Tic-Tac-Toe | Tic-Tac-Toe is a two-player game played on a 3x3 grid. Each player takes turns marking a square with their symbol (X or O), aiming to get three of their symbols in a row, column, or diagonal. To play optimally, prioritize securing the center square and blocking opponent's winning moves. |
| Battleship | Battleship is a two-player game where players hide ships on a grid and take turns guessing their opponent's ship locations. The goal is to sink all of the opponent's ships. To play optimally, start by targeting areas with higher probabilities of containing a ship and strategically target adjacent squares after a hit to maximize efficiency. |
| Connect Four | Connect Four is a two-player game played on a 6x7 grid. Players drop colored discs into columns, aiming to connect four of their own discs in a row, column, or diagonal. To play optimally, prioritize creating your own winning formations while blocking opponent's potential winning moves. |

Table 1: Optimal strategies for playing different games according to GPT-3.5.

| Game | Explanation |
|---|---|
| Tic-Tac-Toe | Play your first X in a corner to maximize opportunities. If the opponent plays in the center, play the opposite corner. Block your opponent's potential winning moves and always look to create a line of three. |
| Battleship | Randomize ship placements and start by targeting the center of the grid. Use a checkerboard pattern for efficient searching. Once a ship is hit, focus on the surrounding squares to determine its orientation and sink it. |
| Connect Four | Start in the center column to maximize opportunities in all directions. Build threats vertically, horizontally, and diagonally, and block the opponent's forming lines. Create multiple threats to force the opponent into a defensive position. |

Table 2: Optimal strategies for playing different games according to GPT-4.

