# OpenReview forum: "Show, Don't Tell: Evaluating Large Language Models Beyond Textual Understanding with ChildPlay"
_NeurIPS.cc/2024/Datasets_and_Benchmarks_Track — Submitted to NeurIPS 2024 Track Datasets and Benchmarks_

### Official Review · Reviewer_DMgg · 2024-06-29

**Rating:** 5
**Confidence:** 4

**Review:**

Overall Review
============
This work is interesting and it is a good reality check for the reasoning abilities of LLMs. However, there are important clarity and quality issues that affect reproducibility and understanding of its conclusions (see below). I cannot recommend it for NeurIPS in its current form but I encourage the authors to go over reviewers' feedback to improve their submission.

Quality
=====
* The idea of using visual games to test the reasoning abilities of LLMs is sound.
* One concern is that the technical quality could be improved, for example it is not clear how many runs were used for Figure 5. For example, in Figure 5.a, if you sum up wins and loses for the first blue bar, it adds up to 113 while other bars sum 100. I could not find this information elsewhere in the text. Overall, I think the experimental setup could be described in more detail, making sure all the necessary information to reproduce the experiments is there.
* Another question I have is whether the poor performance of the LLM on, e.g. tic-tac-toe, is due to playing against a random player. This makes me think of move 78 by Lee Sedol against Alpha Go, which basically broke Alpha Go because it was not expecting this movement. The question is, would the LLM improve if matched against minimax or a better player?
* Also, this work would be more complete if the authors studied the effect of the prompt or the pre-prompt on the model abilities (for example, what if you told the LLM the optimal strategy?) as well as the few-shot learning scenario.

Clarity
=====
While the abstract and introduction are easy to read, and the different games are well explained in the supplementary material, I found a lack of clarity on the experimental section.

* It is not clear how many runs were used for Figure 5. For example, in Figure 5.a, if you sum up wins and loses for the first blue bar, it adds up to 113 while other bars sum 100. I could not find this information elsewhere in the text.
* What does GPT-X **random** mean in Figure 6? Is that the performance of the random player when matched against GPT-X? If so, please clarify because it might be confused with different sampling strategies for GPT.
* Battleship results are in Figure 14 in the Appendix but they are referenced but line 185 states *"Fig. 5 encompasses comparative results from playing Connect-Four, Tic-Tac-Toe,* **and Battleship.**"
* Figure 8, is difficult to parse since there are many subfigures which are not grouped and the caption in not self-contained. I suggest moving the description of each of the subfigures to the caption and visually group them (e.g. a and b together, e and g together, etc).

Originality
========
* As the authors acknolwedge, the idea of testing LLMs reasoning skills on games has been tried in the past, however, this submission explores this on a different angle, putting more emphasis on the visual aspect and adding two additional games. I believe this angle is interesting and the findings are quite surprising.

Significance
=========
* This work is interesting and a good reality check for the research community in LLMs.
* The authors present a set of new benchmarks because existing benchmarks might have been overfitted upon. However, nothing prevents new LLMs to overfit the new proposed benchmark, so it might be shortlived. It would be interesting if the authors generated a third new game but kept it hidden, only providing tools for evaluation.
* In the same spirit, the [ARC-AGI](https://arcprize.org) dataset and challenge, which contains a hidden test set and consists of visual grid patterns that can easily be discretized and fed to an LLM (which currently fail or [perform poorly](https://www.lesswrong.com/posts/Rdwui3wHxCeKb7feK/getting-50-sota-on-arc-agi-with-gpt-4o) on the test set). It would be interesting to see some discussion from the authors on how this submission compares with ARC and what are the different learning that can be extracted from these two different benchmarks.

**Strengths:**

* Interesting approach of using visual games to test LLM reasoning abilities.
* Highlights an interesting and underexplored angle, focusing on the visual aspect of reasoning.
* Adds new games to the benchmark, providing a broader evaluation spectrum.
* Presents surprising and valuable findings that contribute to understanding LLM capabilities.
* Offers a reality check for the research community on the current state of LLMs.
* I was skeptical about the fact that LLMs are bad at tic-tac-toe, so I tried myself on qwen2-72b and the LLM completely failed.

**Additional Feedback:**

* Besides the suggestions above, it would be interesting to see some game logs for all the games.

**Clarity:**

While the abstract and introduction are easy to read, and the different games are well explained in the supplementary material, I found a lack of clarity on the experimental section.

* It is not clear how many runs were used for Figure 5. For example, in Figure 5.a, if you sum up wins and loses for the first blue bar, it adds up to 113 while other bars sum 100. I could not find this information elsewhere in the text.
* What does GPT-X **random** mean in Figure 6? Is that the performance of the random player when matched against GPT-X? If so, please clarify because it might be confused with different sampling strategies for GPT.
* Battleship results are in Figure 14 in the Appendix but they are referenced but line 185 states *"Fig. 5 encompasses comparative results from playing Connect-Four, Tic-Tac-Toe,* **and Battleship.**"
* Figure 8, is difficult to parse since there are many subfigures which are not grouped and the caption in not self-contained. I suggest moving the description of each of the subfigures to the caption and visually group them (e.g. a and b together, e and g together, etc).

**Correctness:**

The claims made in the submission are generally correct, but there are several areas that require improvement for better clarity and reproducibility. The experimental setup lacks detail, such as the number of runs used for Figure 5, leading to confusion about the data presented. The performance against random players may not fully represent the LLM's capabilities, and it would be beneficial to test against stronger opponents and explore the impact of different prompts or pre-prompts on performance. The clarity of figures and captions needs enhancement to ensure they are self-contained and easy to understand. While the originality of focusing on visual reasoning in games is commendable, more comparisons with existing benchmarks like ARC-AGI could strengthen the submission. Overall, the submission is significant but would benefit from more detailed and transparent experimental descriptions.

**Documentation:**

The authors provide the code on github and a detailed README. Given that models are tested against a random agent, and my concerns above about Figure 5, I am not confident that results can be easily reproduced.

**Ethics:**

I do not have any ethical concerns.

**Limitations:**

The authors properly discuss limitations in Section 4.1.

**Opportunities For Improvement:**

Quality
=====
* Clearly state the number of runs used for Figure 5 and ensure consistency in reported values (e.g., first blue bar sums to 113 while others sum to 100).
* Provide a more detailed description of the experimental setup to enable reproducibility.
* Investigate whether the LLM's poor performance on games like tic-tac-toe is due to playing against a random player and consider testing against stronger opponents like minimax.
* Study the effect of the prompt or pre-prompt on the model's abilities and explore few-shot learning scenarios.

Clarity
=====

* Specify the number of runs used for Figure 5 and clarify discrepancies (e.g., the first blue bar summing to 113 vs. 100).
* Explain what GPT-X random means in Figure 6 to avoid confusion with different sampling strategies for GPT.
* Correct the reference to Battleship results being in Figure 14 instead of Figure 5 as mentioned in line 185.
* Improve Figure 8 by grouping subfigures visually and providing a detailed, self-contained caption.

Originality
========

* Highlight the novel angle of emphasizing visual aspects and adding two new games, differentiating this work from previous studies.

Significance
=========

* Discuss the potential for new LLMs to overfit the proposed benchmarks and suggest the creation of a hidden third game for evaluation.
* Compare this work with the ARC-AGI dataset and challenge, discussing different insights and learnings from both benchmarks.

**Relation To Prior Work:**

Raven Progressive Matrices (RPM) [A, B, C] and ARC [D] are two typical benchmarks discussed in the abstract reasoning literature. However they are not mentioned in this work.

[A] Webb, Taylor, Keith J. Holyoak, and Hongjing Lu. "Emergent analogical reasoning in large language models." Nature Human Behaviour 7.9 (2023): 1526-1541.

[B] Barrett, David, et al. "Measuring abstract reasoning in neural networks." International conference on machine learning. PMLR, 2018.

[C] Klein, Balázs, and Kristof Kovacs. "The Performance of Chatgpt and Bing on a Computerized Adaptive Test of Verbal Intelligence." Available at SSRN 4597520.

[D] Chollet, François. "On the measure of intelligence." arXiv preprint arXiv:1911.01547 (2019).

**Summary And Contributions:**

The authors propose a series of text-based visual games to test the reasoning abilities of LLMs. The main appeal of using visual games in ASCII format is to minimize the possibility that the LLM had been trained on them. Concretely, the list of games evaluated in this work is: TIC-TAC-TOE, connect four, Battleship, and two new games on recognizing shapes and rule-based positioning of LEGO pieces. Results show that LLMs including GPT4 are bad at these tasks, even if they are able to provide the optimal strategy to win the game, which puts into question how much of current LLMs' code and language reasoning skills come from overfitting the training set rather than broad generalization.

---

> ### Author Response · Authors · 2024-08-17
> **One concern is that the technical quality could be improved, for example it is not clear how many runs were used for Figure 5. For example, in Figure 5.a, if you sum up wins and loses for the first blue bar, it adds up to 113 while other bars sum 100. I could not find this information elsewhere in the text. Overall, I think the experimental setup could be described in more detail, making sure all the necessary information to reproduce the experiments is there.**
>
> Thank you for your feedback regarding the clarity and detail of the experimental setup. We acknowledge the confusion around Figure 5 and apologize for the oversight. The discrepancy you pointed out, where the first blue bar sums to 113 instead of 100, is indeed an error. It was brought by the fact that we first ran the t=0 (t for temperature) setup 1000 times - this was the first experiment we ran and we expected much more diverse results than what we found. Seeing the repetitive nature of the gameplay at t=0, we ran the remaining conditions 100 times. There was a mistake in the averaging calculation for t=0. We now reran the experiments for t=0 100 times such that no intermediate computations are necessary, and have updated the figure as well in the updated version of the manuscript. Furthermore, we added an algorithmic description to the Appendix regarding the games. The code is open-source so other researchers can go and rerun it already regardless.

---

> ### Author Response · Authors · 2024-08-17
> **Another question I have is whether the poor performance of the LLM on, e.g. tic-tac-toe, is due to playing against a random player. This makes me think of move 78 by Lee Sedol against Alpha Go, which basically broke Alpha Go because it was not expecting this movement. The question is, would the LLM improve if matched against minimax or a better player?**
>
> Unfortunately, the remaining reviewing time is not sufficient to have the LLMs compete against stronger opponents and also perform comprehensive analysis. Most importantly, this experiment has already been done in different studies, both matching GPT-4 against GPT-4 and GPT-4 against minimax (refer to https://ceur-ws.org/Vol-3563/paper_14.pdf for an example). Nevertheless, we agree that this would be a valuable extension of our work when done for all the games we probed. Note that the choice to match the LLM against a random player was based on our supposition that a random player would be easy to beat in these simple games.
> Importantly, we already discuss these results by Davide Liga and Luca Pasetto in our introduction.

---

> ### Author Response · Authors · 2024-08-17
> **Also, this work would be more complete if the authors studied the effect of the prompt or the pre-prompt on the model abilities (for example, what if you told the LLM the optimal strategy?) as well as the few-shot learning scenario.**
>
> We agree that exploring the impact of different prompts on the LLM's performance would be insightful. Additionally, testing the model's abilities in a few-shot learning scenario would help us explore the model’s capacity to learn and adapt to new information. However, we feel that this is an entirely different paper altogether.
>
> Importantly, we already carried out pre-prompting by first prompting each model about optimal play in each game (included in the Appendix B.5) where the models express knowledge of the optimal playing strategies for each game. Technically, therefore, the models know optimal play in these games and they therefore encountered it during training. Hence, it should not be necessary to tell the models the optimal strategy.
> While studying the effect of prompting was outside the scope of our current work (we aimed at a zero-shot experiment), we acknowledge this in our current updated discussion and suggest it as an important direction for future research.

---

> ### Author Response · Authors · 2024-08-17
> **While the abstract and introduction are easy to read, and the different games are well explained in the supplementary material, I found a lack of clarity on the experimental section.**
>
> We thank the reviewer for pointing this out. In the updated version of the manuscript, we have made significant efforts to improve the experimental section to improve clarity.

---

> ### Author Response · Authors · 2024-08-17
> **What does GPT-X random mean in Figure 6? Is that the performance of the random player when matched against GPT-X? If so, please clarify because it might be confused with different sampling strategies for GPT.**
>
> We apologize for the ambiguity in Figure 6 regarding the term "GPT-X random." This indeed refers to the performance of the random player when matched against GPT-X, but we understand how this could be confused with different sampling strategies for GPT. We have revised the caption to clarify this, ensuring that it is clear to the reader that "GPT-X random" refers to the performance of a random player when matched against the specified LLM.

---

> ### Author Response · Authors · 2024-08-17
> **Battleship results are in Figure 14 in the Appendix but they are referenced but line 185 states "Fig. 5 encompasses comparative results from playing Connect-Four, Tic-Tac-Toe, and Battleship."**
>
> Thank you for catching the inconsistency regarding the Battleship results. We have  corrected the reference in line 185 to accurately reflect where the Battleship results are presented. In fact, we have moved the Battleship results from the Appendix to the main body of the text in the next updated version of the manuscript.

---

> ### Author Response · Authors · 2024-08-17
> **Figure 8, is difficult to parse since there are many subfigures which are not grouped and the caption in not self-contained. I suggest moving the description of each of the subfigures to the caption and visually group them (e.g. a and b together, e and g together, etc).**
>
> We agree that Figure 8 could be more clearly presented. We have revised the Figure by removing redundant subfigures and by providing a more detailed and self-contained caption per subfigure.

---

> ### Author Response · Authors · 2024-08-17
> **The authors present a set of new benchmarks because existing benchmarks might have been overfitted upon. However, nothing prevents new LLMs to overfit the new proposed benchmark, so it might be shortlived. It would be interesting if the authors generated a third new game but kept it hidden, only providing tools for evaluation**
>
> Your concern about the risk of overfitting on new benchmarks is valid. While no benchmark can completely prevent future models from overfitting, zero-shot benchmarks are specifically designed to test models without prior training on the task. Training on the task before testing would defeat the purpose of the zero-shot evaluation. Therefore, if researchers choose to train their model on the child-play benchmark and then report their results, they will need to disclose that. For commercial models, we can certainly not prevent this from happening with an openly available benchmark.
> Having said that, we appreciate your suggestion to create a hidden benchmark and will try to do so in the future. However, the time constraints for this paper submission do not give us sufficient time to do this. Needless to say, properly generating a new game dataset requires careful analysis and possibly redesigning the experimental setup, which will be very time-intensive and take substantial effort.

---

> ### Author Response · Authors · 2024-08-17
> **In the same spirit, the ARC-AGI dataset and challenge, which contains a hidden test set and consists of visual grid patterns that can easily be discretized and fed to an LLM (which currently fail or perform poorly on the test set). It would be interesting to see some discussion from the authors on how this submission compares with ARC and what are the different learning that can be extracted from these two different benchmarks.**
>
> We are actually very interested in the ARC-AGI competition, and the similarities were not lost on us even during writing. We appreciate the suggestion. Both our work and ARC-AGI aim to explore the reasoning capabilities of AI systems, albeit through different methodologies. This comparison could help situate our work within the broader context of AI benchmarks and provide additional perspectives on how different tasks can be used to probe the reasoning abilities of LLMs. We also believe that the failure of deep learning based AI systems to solve the ARC problems matches our expectations given our results. Therefore, we added direct references to this dataset and challenges and discussed similarities between them.

---

> ### Comment · Reviewer_DMgg · 2024-08-20
>
> Dear Authors,
>
> Thank you for addressing each of my concerns in your rebuttal. I particularly appreciate the efforts made to improve the clarity of the paper. While some limitations remain due to the constraints of the rebuttal period, I acknowledge the progress made and am willing to raise my score to 5.
>
> DMgg

---

> > ### Author Rebuttal · Authors · 2024-08-27
> >
> > After our last update, we have re-evaluated your comments and we realized that a hidden benchmark can have a significant impact in future benchmarking efforts. Hence, we have implemented, tested, and benchmarked the GPT models on a new hidden benchmark that corresponds to a chemistry game we invented and called Guess-the-SMILES (GtS). This game is a combination of two distinct parts: a Flask web application that interacts with an ASCII generation and evaluation system, and a benchmarking script that tests the performance of different AI models on predicting the established molecular string representation SMILES (Simplified Molecular Input Line Entry System) from ASCII drawings of molecules. We also enable the generation of png images for later use in multi-modal model benchmarking. However, we do not provide the source code and underlying data of the benchmark to prevent future models from overfitting to this benchmark. Notably, users can test this benchmark via a web application interface at: https://child-play.onrender.com/
> >
> > We ran the benchmark 100 times for each temperature setting for both GPT-3.5 and GPT-4 and our findings corroborate our previous results, that GPT models cannot partake in spatial reasoning. This is in spite of comprehensive knowledge of the models about the SMILES representation and chemistry in general, which we assessed comprehensively via prompting. We have added this to the new pdf submission which you can find attached to the new rebuttal.

---

### Official Review · Reviewer_E51U · 2024-07-16
**Review for Paper #2570**

**Rating:** 5
**Confidence:** 4
**Clarity:** Yes.

**Review:**

Please see below for a list of strengths and weaknesses of the paper.

**Strengths**

1. The motivation behind constructing LCL challenges is interesting as we need to test state-of-the-art LLMs outside their learned data distributions and present a fresh and innovative approach to evaluating LLMs beyond linguistic tasks, providing a more holistic assessment of cognitive capabilities.

2. The benchmark study highlights the limitations of current LLMs, such as GPT-3.5 and GPT-4, in strategic gameplay and spatial reasoning tasks, offering valuable insights into their true capabilities and limitations.

**Weaknesses**

1. While the chosen games provide valuable insights, the selection might still be limited in scope as we do not whether the LLMs are learning these tasks according to the specified rules or not.

2. The results show that model performance can vary significantly with different temperature settings, which might complicate the interpretation of the findings and the consistency of the evaluation.

3. Encoding the games via ASCII might limit the complexity and richness of the tasks. More sophisticated representations could potentially offer deeper insights into the models' capabilities.

4. The study is focused on GPT-3.5 and GPT-4, which may limit the generalizability of the findings to other LLMs or AI models. Testing a broader range of models could strengthen the conclusions.

**Strengths:**

Please see the review for more details.

**Additional Feedback:**

N/A

**Correctness:**

Somewhat yes as there are questions about the generalizability of the presented results.

**Documentation:**

Yes.

**Limitations:**

See the review section for more details.

**Opportunities For Improvement:**

1. More detailed captions in figures would be beneficial for the reader. For instance, in Fig. 4, the authors just mention "Minimax vs random player" which does not describe anything about the figure. What do the numbers represent? The Agent represents which LLM? etc.

2. How did we ensure the LLM understood the instructions in Lines 128-131? Further, Validity Testing quantifies the validity of each LEGO construct but doesn't ensure whether the LLM understood the actual reasoning/rules that determine the validity.

3. Lines 226-229: The authors find that In the shape detection tests, GPT-3.5’s performance is close to random chance but GPT-4 successfully identifies shapes with an accuracy of 80% or higher, but do not explain any intuition behind why this may be the case. Are the authors attributing this increased performance to just the size of the models?

4. The authors propose these datasets as benchmarks but only test the GPT-family LLMs. To show the robustness of the benchmark, it would be beneficial if the authors could compare the performance of other LLMs like Llama-family, Mistral-family, etc.

**Relation To Prior Work:**

Yes.

**Summary And Contributions:**

The paper aims to explore the cognitive capabilities of large language models (LLMs) like GPT-3.5 and GPT-4 beyond traditional linguistic tasks. In particular, the author introduces a novel evaluation approach using a suite of games, named ChildPlay, which includes Tic-Tac-Toe, Connect Four, Battleship, LEGO Connect Language (LCL), and a game of shapes, which are designed to test strategic thinking, decision-making, spatial reasoning, and general cognitive abilities. The findings indicate that despite proficiency in standard benchmarks, both GPT models show limited abilities in strategic gameplay and spatial reasoning, revealing a significant blind spot in current LLM evaluations.

---

> ### Author Response · Authors · 2024-08-17
> **While the chosen games provide valuable insights, the selection might still be limited in scope as we do not (know) whether the LLMs are learning these tasks according to the specified rules or not.**
>
> Thank you for raising the point about the scope of the chosen games and whether the LLMs are truly learning the tasks according to the specified rules. We acknowledge that our benchmarks are limited in the sense that they primarily rely on the models following explicit rules provided through prompts. However, our approach aims to assess how well these models can internalize and apply these rules in a structured environment. To address your concern, we have clarified in the manuscript that while our benchmarks attempt to probe the model's understanding indirectly through game performance, gameplay is inherently limited by the model's reliance on probabilistic patterns rather than genuine rule-based reasoning, something that is contentious in the field at the moment and that we attempted to probe. We also now make it clear that the rules are explicitly taught to the models through prompting, which we believe is a critical component of evaluating how well LLMs can generalize from given instructions. However, we recognize that this does not fully guarantee that the models are learning and applying these rules as a human would, but that fact is what we aimed at demonstrating. We also include in the Appendix a section showing that when prompted about rules and optimal play, both models are capable of expressing detailed valid knowledge in that regard.

---

> > ### Comment · Reviewer_E51U · 2024-08-24
> >
> > Thank you for your detailed rebuttal response. Considering the rating from other reviewers, I would like to retain my score.

---

> > > ### Author Response · Authors · 2024-08-24
> > >
> > > Dear Reviewer,
> > > Thank you for acknowledging our detailed response. We wonder whether there any points from your concerns that we have not addressed sufficiently in our response. Hence, we would appreciate your comments on our rebuttal.

---

> ### Author Response · Authors · 2024-08-17
> **The results show that model performance can vary significantly with different temperature settings, which might complicate the interpretation of the findings and the consistency of the evaluation.**
>
> You are correct in noting that varying temperature settings can lead to significant differences in model performance, which might complicate the interpretation of results. While we understand this could introduce some variability, temperature is known to influence the diversity of outputs by controlling the randomness of the model’s predictions and our intention was to probe exactly that influence.
> In response to your concern, we have added a more detailed explanation of how temperature settings affect model behavior in the main text. This explanation will help contextualize the results, making it clearer that temperature adjustments are a deliberate tool used to explore the range of possible model behaviors. We now also discuss how this variability impacts the consistency of our evaluation and the interpretation of the results.

---

> ### Author Response · Authors · 2024-08-17
> **Encoding the games via ASCII might limit the complexity and richness of the tasks. More sophisticated representations could potentially offer deeper insights into the models' capabilities.**
>
> Our choice of representation was driven by the simplicity and accessibility it provides, allowing for straightforward interaction between the models and the tasks. The point we are trying to make is that if the LLM cannot solve the simplest version of the game, then it will most likely not be able to solve anything more complex either. There is one interesting question here though, whether using a different representation for the same model has an impact on the performance. It possibly has. However, the games chosen are fully-observable in the sense that any information necessary to play optimally is present in the board game at every state. Regardless,  we appreciate your feedback on the use of ASCII to encode the games and we now include a discussion in the manuscript about the potential impact of using more sophisticated representations, such as more complex symbolic representations, on the models' performance. While our current focus was on simple games with straightforward rules, we acknowledge that exploring alternative representations could offer deeper insights into the models' capabilities. Nevertheless, we believe that studying the influence of representations in a systematic way is outside the scope of this work.

---

> ### Author Response · Authors · 2024-08-17
> **The study is focused on GPT-3.5 and GPT-4, which may limit the generalizability of the findings to other LLMs or AI models. Testing a broader range of models could strengthen the conclusions.**
>
> We acknowledge that our study only included GPT-3.5 and GPT-4, and we agree that expanding the range of models tested might strengthen our findings. However, we believe that benchmarking GPT-3.5 and GPT-4 is already sufficient to make our claims. Based on other benchmarks, at the time we performed this work, no other model surpassed the GPT series in terms of their capacity to follow commands or rules.
> In the current updated  version we test other models in a softer manner and include the results in the appendix. This includes prompting babbling models and playing fulls games against the better competitors (Mistral, Claude, and Gemini).

---

> ### Author Response · Authors · 2024-08-17
> **More detailed captions in figures would be beneficial for the reader. For instance, in Fig. 4, the authors just mention "Minimax vs random player" which does not describe anything about the figure. What do the numbers represent? The Agent represents which LLM? etc.**
>
> We agree with your suggestion to provide more detailed captions in our figures. For example, in Figure 4, the caption has been revised to include a more comprehensive explanation of what the numbers represent, which model is referred to as the 'Agent' (it is the minimax algo) and the specific dynamics being depicted (moves per game over all games).

---

> ### Author Response · Authors · 2024-08-17
> **How did we ensure the LLM understood the instructions in Lines 128-131? Further, Validity Testing quantifies the validity of each LEGO construct but doesn't ensure whether the LLM understood the actual reasoning/rules that determine the validity.**
>
> Your concern about how we ensure the LLMs understood the instructions is valid. The lines you referenced (128-131) describe how we prompt the models with the rules of the games. While we quantify the validity of their outputs in terms of LEGO constructs, this does indeed not directly ensure that the models fully grasp the reasoning or rules behind the tasks. However, we believe that the rules are simple enough and if we suppose that GPT can understand English then it should be able to understand the rules. This is an important point and is essentially what we are trying to probe, whether GPT or comparable LLMS genuinely understand. If GPT fails at the game we now conclude that it does not understand the rules even though it can reproduce them when prompted. Our method is indirect because we cannot trust the LLM to tell us if it understood something or not.

---

> ### Author Response · Authors · 2024-08-17
> **Lines 226-229: The authors find that In the shape detection tests, GPT-3.5’s performance is close to random chance but GPT-4 successfully identifies shapes with an accuracy of 80% or higher, but do not explain any intuition behind why this may be the case. Are the authors attributing this increased performance to just the size of the models?**
>
> Thank you for highlighting the need for a deeper explanation of the performance differences between GPT-3.5 and GPT-4 in the shape detection tests. The significant performance gap, where GPT-4 outperforms GPT-3.5, is indeed intriguing and warrants further discussion. It is problematic that OpenAI does not disclose most relevant information regarding training methodology, model architecture, or training data content. We do believe that LLM size matters and some think it is the only thing that matters. However, more importantly, the data used for training might have contained shape matrices similar or even equal to the ones we used. It is not unthinkable that someone would have made use of such ASCII objects. Accordingly, in the revised manuscript, we add these reasons as an explanation of this performance discrepancy.

---

> ### Author Response · Authors · 2024-08-17
> **The authors propose these datasets as benchmarks but only test the GPT-family LLMs. To show the robustness of the benchmark, it would be beneficial if the authors could compare the performance of other LLMs like Llama-family, Mistral-family, etc.**
>
> In the current updated  version we test other models in a softer manner and include the results in the appendix. This includes prompting babbling models and playing fulls games against the better competitors (Mistral, Claude, and Gemini).

---

### Official Review · Reviewer_MtWS · 2024-07-27
**Interesting LLM evaluation with games**

**Rating:** 5
**Confidence:** 3

**Review:**

This paper provides a new suite of non-language-based games to evaluate LLMs, and present some blind spots of current LMs which were  rarely covered in previous studies.

Pros:
1. The paper addresses an important issue in evaluating LLMs by proposing non-language-based tasks that minimize dataset contamination and focus on assessing reasoning capabilities.
2. The authors introduce novel tasks, such as the LEGO Connect Language (LCL) and the game of Shapes, which challenge LLMs to demonstrate spatial reasoning and pattern recognition abilities.
3. The experiments are well-designed and provide insights into the limitations of current LLMs in tasks requiring precise strategic reasoning and rule adherence.

Cons:
1. It's unclear to me whether the performance on these "games" can reflect the realistic performance in everyday use, which further questions the value of this work. For example, does high score in Battleship mean the model can perform well in decision making? If not, to what extent this suite can serve as a reference for model developers or users?
2. The authors could provide more information on the scalability of the proposed tasks and their potential to be incorporated into existing benchmarks.
3. The paper could benefit from a more detailed discussion of the implications of the results for the development of LLMs and their potential applications.

In general, although the paper has merits the first con is the deal breaker. I suggest the author better discuss the connection with real-world applications and the possible takeaways from running the evaluations.

**Strengths:**

See the above "Pros" in the review section.

**Additional Feedback:**

No.

**Clarity:**

The paper was mostly well written. Some figures are be further improved. For examples, the words in Figure 8 and 9 are too small to see clearly.

**Correctness:**

The authors properly discussed the limitation and the models they used. The experiments are correctly set.

**Documentation:**

Yes, the authors will open sourced their dataset on Github.

**Ethics:**

No.

**Limitations:**

1. The paper acknowledges that the use of binary (win/loss) outcomes for games may exaggerate perceived capabilities or weaknesses of LLMs.
2. The experiments are limited to two specific LLMs (GPT-3.5 and GPT-4), and the generalizability of the findings to other LLMs is not explored.
3. The paper does not provide error bars or other statistical significance measures for the experimental results, which could help in interpreting the findings.

**Opportunities For Improvement:**

See the review section.

**Relation To Prior Work:**

As far as I know there are many text-based games projects before, such as: https://www.microsoft.com/en-us/research/project/textworld.

**Summary And Contributions:**

The paper introduces ChildPlay, a suite of non-language-based games such as Tic-Tac-Toe, Connect Four, Battleship, LEGO Connect Language (LCL), and the game of Shapes, to assess the reasoning, strategic capabilities, symbolic reasoning, and pattern recognition abilities of large language models (LLMs) beyond traditional linguistic modalities. The authors argue that these games provide structured environments with clear success criteria, making them suitable for evaluating the strategic thinking, planning, and long-term decision-making capabilities of LLMs.

---

> ### Author Response · Authors · 2024-08-17
> **It's unclear to me whether the performance on these "games" can reflect the realistic performance in everyday use, which further questions the value of this work. For example, does high score in Battleship mean the model can perform well in decision making? If not, to what extent this suite can serve as a reference for model developers or users?**
>
> We appreciate your comments regarding the applicability of the game-based tasks to real-world performance. We also understand your concern about the relevance of these tasks in assessing a model's everyday decision-making capabilities. Our study aims to explore the emergent cognitive capabilities of LLMs through indirect testing via structured games like Battleship, which require strategic thinking and spatial reasoning. While a high score in Battleship may not directly translate to general decision-making performance, the underlying reasoning skills assessed through these games can offer valuable insights into the model's problem-solving abilities. Additionally, the key point of these games is that the quality of the moves selected by the model can readily be evaluated objectively as the games are simple enough.
> The key takeaway from our work is not solely about the models' performance in specific games, but rather about understanding the extent to which these models can generalize and apply reasoning in novel contexts. Therefore, these findings have implications for the broader field of AI, particularly in the development of models truly capable of reasoning, or of adaptive and flexible thinking. We believe this research provides insight into how LLMs handle tasks that require higher-order reasoning, which is crucial for more advanced applications and, therefore, assesses the current state of the field in that respect.
> In a more practical sense, regardless of the theoretical relevance, in case GPT or other LLMs cannot reason appropriately regarding novel data, then we also should not ask it to reason about novel data in the context of standard applications (e.g. when using chatGPT). This is something we consider critical, especially in a scientific context. We believe that, currently, many users (both scientists and non-scientists) are unaware of this and are likely biased towards believing an LLM’s output because it reads genuine and seems believable.  We think that our benchmarks are well-suited to illustrate these pitfalls as we expect essentially anyone that uses LLMs to be able to play the games in ChildPlay. Accordingly, we believe that seeing how current LLMs fail to play these simple games will make users more cautious about the answers and question their correctness.

---

> ### Author Response · Authors · 2024-08-17
> **The authors could provide more information on the scalability of the proposed tasks and their potential to be incorporated into existing benchmarks.**
>
> We appreciate your suggestion to provide more details on the scalability of our proposed tasks. The suite of games we employed is designed to be computationally both simple and feasible. Hence they  can be scaled to evaluate different model sizes and configurations. For example, the benchmarks can be executed relatively quickly, typically requiring only a few hours to complete across all tasks for a given model and we provide this information in the paper.
> Furthermore, we have already proposed this benchmark suite for inclusion in the BIG-bench initiative, which aims to standardize challenging benchmarks for LLMs. While this information may not be necessary to be added to the manuscript itself, as we do not know yet whether the benchmarks will be included in BIG-bench, it highlights the relevance and potential adoption of our tasks in broader benchmarking efforts.

---

> ### Author Response · Authors · 2024-08-17
> **The paper could benefit from a more detailed discussion of the implications of the results for the development of LLMs and their potential applications.**
>
> We agree that our findings have significant implications for the development of LLMs. Specifically, our results suggest that while LLMs may exhibit some reasoning abilities, these are likely not emergent (yet) except in that they follow from the designed probabilistic outputs enabled by the LLM algorithm. In other words, any reasoning chain present in the output is likely directly reducible to some reasoning chain present in training data or mixtures thereof. Current LLMs still face challenges in consistently applying these abilities across different tasks, especially in scenarios that require multi-step decision-making.
> In the revised manuscript, we have expanded the discussion section to include a more comprehensive analysis of how these results can inform the future development of LLMs. This includes considerations for enhancing the models' ability to generalize across tasks and the importance of integrating reasoning capabilities into LLM architectures.

---

> ### Author Response · Authors · 2024-08-17
> **The experiments are limited to two specific LLMs (GPT-3.5 and GPT-4), and the generalizability of the findings to other LLMs is not explored.**
>
> We acknowledge that our study only included GPT-3.5 and GPT-4, and we agree that expanding the range of models tested might strengthen our findings. However, we believe that benchmarking GPT-3.5 and GPT-4 is already sufficient to make our claims. Based on other benchmarks, at the time we performed this work, no other model surpassed the GPT series in terms of their capacity to follow commands or rules.
> Regardless, in the current updated version we test other models in a softer manner and include the results in the appendix. This includes prompting babbling models and playing full games against the better competitors (Mistral, Claude, and Gemini).

---

> ### Author Response · Authors · 2024-08-17
> **The paper does not provide error bars or other statistical significance measures for the experimental results, which could help in interpreting the findings.**
>
> We recognize the importance of including error bars or other measures of statistical significance to aid in the interpretation of experimental results - this was a lapse on our part. We have incorporated error bars in the relevant figures in the revised manuscript to provide a clearer understanding of the variability in the models' performance.

---

### Official Review · Reviewer_4ccX · 2024-07-31
**A new benchmark to assesse the cognitive abilities of GPT-3.5 and GPT-4**

**Rating:** 4
**Confidence:** 4
**Correctness:** As mentioned above, the experimental …

**Review:**

Pros:

- This work has produced a novel suite of datasets to evaluate the performance of large language models on non-linguistic tasks and spatial reasoning abilities.
- The author has made the relevant tools and experimental data available, which will be beneficial for research in the related field.

Concerns:

- The introduction of each letter is missing in line 128 of the text. Additionally, it would be beneficial to include some sample data.
- The markings in Figure 4 are not clear enough. Does 'agent' refer to the minimax algorithm? Additionally, the heatmap seems unrelated to the purpose and lacks discussion in the text.
- According to the design in the text, the results of Battleship should be one of the main results. However, Figure 14 is placed in the appendix. The presentation of the main results may need to be more intuitive. I suggest using a table to display the win rates of each model.
- The font in the appendix figures is too small and the labels are unclear. In Figures 10, 11, 15, and 16, each subfigure in the heatmaps does not indicate which player it corresponds to. Additionally, as previously mentioned, the role of these heatmaps in the text is unclear. The text lacks discussion about these heatmaps, and they do not support any points made in the text.
- The citation of Figure 16 in line 189 seems unrelated to the claim.
- In line 193, why does the Random Player exhibit a normal distribution? In my understanding, if the moves are random, it should approximate a uniform distribution. Additionally, the citation of Figure 12 here is also incorrect.
- In line 200, the author suggests that higher temperature broadens the explored moves within the models’ strategies, thus reducing missed wins and blocks. Could this be due to increased randomness leading to more incorrect moves, which in turn causes the game to end more quickly? I believe it is necessary to provide statistics on the probability of incorrect moves occurring and the average number of steps when they occur as the temperature increases.
- The text only tested GPT-3.5 and GPT-4. Are there any results from testing on other models?
- Why is the heatmap for Connect-Four presented in a single column, rather than in a grid format like Tic-Tac-Toe?
- Overall, both models perform poorly across multiple games set in the text. Could this be due to the task format being unsuitable or overly difficult for the current LLMs, making it hard for them to understand the tasks? I believe the author may need to include methods such as Chain-of-Thought (CoT) or Few-Shot learning for testing. Otherwise, if all LLMs fail on this benchmark, it may lack practical value for evaluation.

**Strengths:**

This work provides a novel test dataset, which could be helpful for evaluating the reasoning abilities of LLMs and mitigating the issue of current test datasets being contaminated.

**Additional Feedback:**

Included in the review section.

**Clarity:**

Overall, the article is clear, but some figures and explanations need improvement.

**Documentation:**

Yes

**Limitations:**

Yes, the author has expressed some limitations, but the aforementioned weaknesses also need to be addressed.

**Opportunities For Improvement:**

The experimental design and presentation in the text need improvement. More comprehensive experiments may be necessary to validate the quality and practical value of the dataset.

**Relation To Prior Work:**

Yes

**Summary And Contributions:**

The paper presents a new benchmark to assesse the cognitive abilities of GPT-3.5 and GPT-4 using a suite of non-linguistic games. These include Tic-Tac-Toe, Connect Four, Battleship, LEGO Connect Language (LCL), and a shapes game, all encoded in ASCII to evaluate strategic thinking, decision-making, and spatial reasoning.

---

> ### Author Response · Authors · 2024-08-17
> **Answering Reviewer 4ccX31 - The introduction of each letter is missing in line 128 of the text. Additionally, it would be beneficial to include some sample data.**
>
> A. 1. This is a good point. We decided to only have a verbal definition of LCL at this point and remove the formal definition of LCL in line 128 of the main text and instead refer to the appendix in the next version of the manuscript. As suggested, we also added sample data for the experiments of the LCL section. Additionally, we added two tables describing the data for all the games in the Appendix.

---

> ### Author Response · Authors · 2024-08-17
> **The markings in Figure 4 are not clear enough. Does 'agent' refer to the minimax algorithm? Additionally, the heatmap seems unrelated to the purpose and lacks discussion in the text.**
>
> Thank you for pointing out the issues with Figure 4. We acknowledge that the markings could have been clearer. To clarify, 'agent' indeed refers to the minimax algorithm. We added an explanatory description that we hope is more intuitive to the text after the image and explained the term 'agent' in the figure caption.
> Regarding the heatmap, we agree that its role in the current version of the manuscript may seem disconnected from the overall discussion. We now provide a more thorough explanation of the heatmap's relevance in the main text, specifically how it relates to the agent's decision-making process and its impact on the game's outcome. We did this in the discussion section.

---

> ### Author Response · Authors · 2024-08-17
> **According to the design in the text, the results of Battleship should be one of the main results. However, Figure 14 is placed in the appendix. The presentation of the main results may need to be more intuitive. I suggest using a table to display the win rates of each model.**
>
> We agree that the Battleship results should be featured more prominently in the main text, given their importance in our study. We appreciate your suggestion to use a table to display the win rates of each model, which would help with communicating our findings. In our next updated version, which we aim to upload next week, we have moved the relevant figure (currently Figure 14) from the Appendix to the main body and supplement the board games with a table that clearly presents the win rates across the different models.

---

> ### Author Response · Authors · 2024-08-17
> **The font in the appendix figures is too small and the labels are unclear. In Figures 10, 11, 15, and 16, each subfigure in the heatmaps does not indicate which player it corresponds to. Additionally, as previously mentioned, the role of these heatmaps in the text is unclear. The text lacks discussion about these heatmaps, and they do not support any points made in the text.**
>
> We appreciate your feedback on the figures in the Appendix, particularly regarding the font size and the clarity of the labels. We have revised these figures to address these concerns. Specifically, we have increased the font size and added an explanation of the heatmaps, indicating which player they correspond to (the left column corresponds to the GPT models and the right column corresponds to the random player). Additionally, in this updated version we have expanded the discussion of these heatmaps in the text, ensuring that their role and relevance is articulated. The heatmaps are insightful because they reveal a certain bias in move strategies for GPT-3.5 and GPT-4 while showing the uniformity of the random player’s moves.

---

> ### Author Response · Authors · 2024-08-17
> **The citation of Figure 16 in line 189 seems unrelated to the claim.**
>
> We acknowledge the errors in the citations and appreciate your careful reading. The citation on line 189 should indeed reference Figure 14, not Figure 16. We have corrected this in the next updated version.

---

> ### Author Response · Authors · 2024-08-17
> **In line 193, why does the Random Player exhibit a normal distribution? In my understanding, if the moves are random, it should approximate a uniform distribution. Additionally, the citation of Figure 12 here is also incorrect.**
>
> Regarding the distribution of the random player's moves, your observation is correct. The expected distribution is indeed uniform, given the random nature of the player's moves. We have revised our explanation in the manuscript to reflect this and corrected the citation of Figure 12.

---

> ### Author Response · Authors · 2024-08-17
> **In line 200, the author suggests that higher temperature broadens the explored moves within the models’ strategies, thus reducing missed wins and blocks. Could this be due to increased randomness leading to more incorrect moves, which in turn causes the game to end more quickly? I believe it is necessary to provide statistics on the probability of incorrect moves occurring and the average number of steps when they occur as the temperature increases.**
>
> We appreciate your comments on the relationship between temperature and game dynamics. While we initially hypothesized that higher temperatures would broaden the range of explored moves, thus reducing missed opportunities, you raise a valid point regarding the potential for increased randomness to lead to more incorrect moves. To address this, we analyzed the heatmaps more carefully. It becomes clear that GPT-3.5, for example, constantly loses at Tic-Tac-Toe from playing in the middle horizontal row, regardless of there being pieces or not. In this case, temperature makes absolutely no difference. Therefore, the effects of temperature increase are not absolute across the board.

---

> ### Author Response · Authors · 2024-08-17
> **The text only tested GPT-3.5 and GPT-4. Are there any results from testing on other models?**
>
> We acknowledge that our study only included GPT-3.5 and GPT-4, and we agree that expanding the range of models tested might strengthen our findings. However, we believe that benchmarking GPT-3.5 and GPT-4 is already sufficient to make our claims. Based on other benchmarks, at the time we performed this work, no other model surpassed the GPT series in terms of their capacity to follow commands or rules.
> In the current updated  version we test other models in a softer manner and include the results in the appendix. This includes prompting babbling models and playing full games against the better competitors (Mistral, Claude, and Gemini).

---

> ### Author Response · Authors · 2024-08-17
> **Why is the heatmap for Connect-Four presented in a single column, rather than in a grid format like Tic-Tac-Toe?**
>
> This is due to the difference in rules of Connect-Four and Tic-Tac-Toe. The move space in Tic-Tac-Toe corresponds to the full grid and the player selects a specific board square. In contrast, in Connect-Four it is a series of 7 columns. By definition, the player can only select a column and each piece piles on top of each other within a given column. Accordingly, this was our conceptual and algorithmic representation of the two games.

---

> ### Author Response · Authors · 2024-08-17
> **Overall, both models perform poorly across multiple games set in the text. Could this be due to the task format being unsuitable or overly difficult for the current LLMs, making it hard for them to understand the tasks? I believe the author may need to include methods such as Chain-of-Thought (CoT) or Few-Shot learning for testing. Otherwise, if all LLMs fail on this benchmark, it may lack practical value for evaluation.**
>
> Your concerns about the overall performance of the models across multiple games are well-taken. As discussed in our introduction, our aim was to explore the emergent cognitive capabilities of LLMs in tasks they were not explicitly trained on, with a focus on zero-shot learning. We agree that this approach inherently involves a higher level of difficulty, which might limit the models' performance.
> Nevertheless, we want to emphasize that the observed performance of the LLMs, though far from perfect, is far from complete failure either and not negligible at all. Therefore, we would argue that the benchmark difficulty is actually at the right level to measure performance differences between current LLMs and, therefore, offers valuable insights into the current state of LLMs. Yet, we acknowledge that the task format might be challenging for the models, and we incorporate a discussion of this in the limitations and future work sections. Specifically, we now suggest exploring methods such as Chain-of-Thought (CoT) prompting or Few-Shot learning in future studies to better assess LLM capabilities under different conditions.

---

### Author Rebuttal · Authors · 2024-08-17

Dear reviewers, we don't seem to be able to edit the pdf submissions, as such we attach the reviewed manuscript as well as the supplementary material here. Furthermore, all reviewer comments have been addressed through official comments under each review. In case we are making a mistake regarding how we attach our revised pdf, we would appreciate any assistance by the area chairs. Thank you.

---

### Author Rebuttal · Authors · 2024-08-27

Dear reviewers,
We created an updated version of our manuscript which is attached to this post. In response to one of the reviewers, the main change is the inclusion of a completely new hidden benchmark in the spirit of all the other benchmarks. We provide API access for this benchmark but do not share the source code and underlying data. Additionally, we improved clarity and writing throughout the entire text and increased the quality of figures.

We believe to have now addressed all the main comments of the reviewers that would not require redoing the entire work from scratch. To summarize all our main changes:
1. We have corrected the image reference mistakes.
2. We have increased the sizes of the images.
3. We added missing error bars.
4. We added proper labeling and descriptions of the images
5. We added results from experiments with many other LLMs
6. We created a hidden benchmark
7. We discussed our work in the light of ARC-AGI and compared results.
8. We moved relevant parts such as the Battleship results from the Appendix to the main text.
9. We added a discussion and results about LLMs competing against a strong opponent compared to a random opponent
10. We included a discussion of the effect of pre-prompting
11. We improved clarity and writing throughout the entire manuscript, especially in the experimental section.

We look forward to further feedback from all the reviewers.

---

### Decision · Program_Chairs · 2024-09-26

**Decision:**

Reject

**Comment:**

The paper aims to explore the cognitive capabilities of large language models (LLMs) like GPT-3.5 and GPT-4 beyond traditional linguistic tasks. In particular, the author introduces a novel evaluation approach using a suite of games, named ChildPlay, which includes Tic-Tac-Toe, Connect Four, Battleship, LEGO Connect Language (LCL), and a game of shapes, which are designed to test strategic thinking, decision-making, spatial reasoning, and general cognitive abilities. The findings indicate that despite proficiency in standard benchmarks, both GPT models show limited abilities in strategic gameplay and spatial reasoning, revealing a significant blind spot in current LLM evaluations.

The reviewers gave low overall scores and the authors made concrete suggestions for how to address those concerns.  We discussed this one at length, we like the idea and approach -- but the final decision is more about how it compares with other papers this year.  The track had 2x more submissions than we expected making it difficult to justify accepting a paper that requires more work over papers that had it all together.  It probably comes as no consolation but this was the highest ranked rejected paper - good luck and we hope to see this published in the coming year.